# Endovascular Treatment of ICAS Patients: Targeting Reperfusion Rather than Residual Stenosis

**DOI:** 10.3390/brainsci12080966

**Published:** 2022-07-22

**Authors:** Tingyu Yi, Alai Zhan, Yanmin Wu, Yimin Li, Xiufen Zheng, Dinglai Lin, Xiaohui Lin, Zhinan Pan, Rongcheng Chen, Mark Parsons, Wenhuo Chen, Longting Lin

**Affiliations:** 1Cerebrovascular and Neuro-Intervention Department, Zhangzhou Affiliated Hospital of Fujian Medical University, Zhangzhou 363000, China; siyuyufen@163.com (T.Y.); minmindoc@163.com (Y.W.); zxf5860@163.com (X.Z.); lindinglai1@163.com (D.L.); linxh@foxmail.com (X.L.); m18050700089@163.com (Z.P.); crc337617@163.com (R.C.); 2Radiology Department, Zhangzhou Affiliated Hospital of Fujian Medical University, Zhangzhou 363000, China; zhanalai@163.com (A.Z.); bluecorn324115@163.com (Y.L.); 3Department of Neurology and Medicine, Royal Melbourne Hospital, University of Melbourne, Melbourne, VIC 3050, Australia; mark.parsons@unsw.edu.au

**Keywords:** atherosclerotic, residual stenosis, reocclusion, endovascular treatment

## Abstract

Background and Purpose: Previous studies showed that acute reocclusion after endovascular therapy is related to residual stenosis. However, we observed that reperfusion status but not residual stenosis severity is related to acute reocclusion. This study aimed to assess which factor mention above is more likely to be associated with artery reocclusion after endovascular treatment. Methods: This study included 86 acute ischemic stroke patients who had middle cerebral artery (MCA) atherosclerotic occlusions and received endovascular treatment within 24 h of a stroke. The primary outcomes included intraprocedural reocclusion assessed during endovascular treatment and delayed reocclusion assessed through follow-up angiography. Results: Of the 86 patients, the intraprocedural reocclusion rate was 7.0% (6/86) and the delayed reocclusion rate was 2.3% (2/86). Regarding intraprocedural occlusion, for patients with severe residual stenosis, patients with successful thrombectomy reperfusion showed a significantly lower rate than unsuccessful thrombectomy reperfusion (0/30 vs. 6/31, *p* = 0.003); on the other hand, for patients with successful thrombectomy reperfusion, patients with severe residual stenosis showed no difference from those with mild to moderate residual stenosis in terms of intraprocedural occlusion (0/30 vs. 0/25, *p* = 1.00). In addition, after endovascular treatment, all patients achieved successful reperfusion. There was no significant difference in the delayed reocclusion rate between patients with severe residual stenosis and those with mild to moderate residual stenosis (2/25 vs. 0/61, *p* = 0.085). Conclusion: Reperfusion status rather than residual stenosis severity is associated with artery reocclusion after endovascular treatment. Once successful reperfusion was achieved, the reocclusion occurrence was fairly low in MCA atherosclerosis stroke patients, even with severe residual stenosis.

## 1. Introduction

Endovascular therapy (EVT) has become a routine practice for acute ischemic stroke caused by large vessel occlusion in highly specialized centers with dedicated stroke units [1,2]. However, EVT procedures do not always lead to good clinical outcomes. One of the reasons is reocclusion of the targeted artery after procedure. The incidence of postoperative acute reocclusion of treated arteries has been reported to range from 3 to 9% [3], and the incidence has been shown to be higher in cases of intracranial atherosclerosis (ICAS)-related occlusion, especially in cases with high residual stenosis [4], because insufficient blood flow caused by high residual stenosis through the target artery leads to acute thrombus formation [4,5,6]. Therefore, if there is sufficient blood flow through the occluded artery affected by ICAS, the incidence of acute thrombosis after endovascular therapy can be relatively low. Furthermore, in our clinical practice, we observed that unsuccessful reperfusion as defined by mTICI < 2b rather than residual stenosis severity was related to artery reocclusion in ICAS patients. We hypothesized that the incidence of acute target arterial reocclusion would be low if mTICI ≥ 2b reperfusion was achieved in patients with MCA ICAS-related occlusion even with high degree of residual stenosis after endovascular therapy. Therefore, our study aimed to assess whether reocclusion was determined by reperfusion status or residual stenosis severity in ICAS patients.

## 2. Methods

### 2.1. Study Design

This retrospective study included the following two-step analysis: (1) Step 1: intraprocedural reocclusion analysis during EVT; (2) Step 2: delayed reocclusion analysis through follow-up angiographic images. Intraprocedural reocclusion was defined as reocclusion during the EVT procedure. Delayed reocclusion was assessed by examining follow-up (2–7 days) angiographic images, which was defined as a sudden cutoff without a distal flow void in magnetic resonance angiography or without the presence of distal flow in computed topography angiography. Reocclusion were evaluated by 2 independent raters (Z.A.L. and L.Y.M).

For Step 1 (analysis of intraprocedural reocclusion), the patients were divided into 3 groups based on the angiographic results during the thrombectomy procedure: Group 1a: successful thrombectomy reperfusion (TICI 2b or 3) + mild to moderate stenosis; Group 2a: successful thrombectomy reperfusion + severe stenosis; and Group 3a: no successful thrombectomy reperfusion + severe stenosis. For Step 2 (analysis of delayed reocclusion), the patients were divided into two groups based on the final angiographic result at the end of the EVT: Group 1b: successful reperfusion + mild-to-moderate stenosis; Group 2b: successful reperfusion + severe stenosis.

### 2.2. Patients

From our prospective registry database, acute ischemic stroke patients admitted between January 2015 and July 2019 were retrospectively reviewed. Patients with anterior circulation stroke and who received EVT within 24 h of stroke onset were selected. Further inclusion criteria were as follows: (1) middle cerebral artery occlusion (tandem occlusion was not included); (2) a diagnosis of ICAS; (3) successful reperfusion that was characterized by a modified thrombolysis in cerebral ischemia (mTICI) Grade 2b to 3 at the end of EVT. Patients with arterial fibrillation that could easily cause cardiac embolism [7] were excluded from this study. Further exclusion criteria were as follows: (1) partial baseline occlusion, (2) no follow-up imaging, and (3) poor imaging quality. The study was approved by the institution’s ethical committee.

### 2.3. Endovascular Procedures

Thrombectomy with stent retrieval was the first endovascular strategy, with emergent angioplasty and/or a stent as a rescue treatment after 1–2 passes of thrombectomy. If successful thrombectomy reperfusion was achieved and could be maintained for more than 20 min, the endovascular therapy was terminated.

### 2.4. Tirofiban Administration and Antiplatelet Regime

A loading dose (10 μg/kg) of a glycoprotein IIb/IIIa inhibitor (tirofiban) was administered intravenously for 3 min once ICAS was considered as the cause of the stroke during the EVT procedure (prior to attempting thrombectomy), then an infusion of tirofiban at 0.1–0.15 μg/kg/min was administered, and this infusion was continued for 12–36 h after the EVT operation [8]. If no brain hemorrhages were detected by the CT scan performed at least 12 h after the operation, a loading dose of aspirin (100 mg/day) plus clopidogrel (300 mg/day), followed by a dose of aspirin (100 mg/day) plus clopidogrel (75 mg/day) for at least 3 months was administered to both patients who received stenting and those who did not.

### 2.5. Definitions of ICAS

ICAS was suspected once the following signs were observed on the first run of DSA: the appearance of a tapered sign [9] or/and significant fixed focal stenosis (>50%) at the site of occlusion during endovascular treatment [10] or/and the phenomenon of “microcrater first-pass effect” [11], or/and a positive stent-unsheathed effect [12]. In addition, a subgroup of patients was scanned by follow-up high-resolution magnetic resonance imaging to confirm the definition.

### 2.6. Evaluation of Angiographic Images

All angiographic classifications, including the grade of collateral flow, the degree of residual stenosis, and the reperfusion score, were evaluated by the two independent raters who rated the reocclusion status (Z.A.L. and L.Y.M). Interobserver disagreements were resolved by consensus.

The baseline grade of collateral flow was evaluated according to the American Society of Interventional and Therapeutic Neuroradiology/Society of Interventional Radiology Collateral Flow Grading System (ASTRIN) through pretreatment angiography. According to this angiographic scale, collateral flow can be classified into Grades 0 to 4 according to the completeness and rapidity of collateral filling in a retrograde manner [13].

The reperfusion statuses was measured using the Thrombolysis in Cerebral Infarction (TICI) scale [13]. An mTICI of 2b-3 was classified as successful reperfusion.

The degree of stenosis was measured by the Warfarin–Aspirin Symptomatic Intracranial Disease Study (WASID) method. The degree of stenosis was classified as mild to moderate (<70%) or high (70–99%) [14].

### 2.7. Outcomes

The primary outcome for Step 1 of the analysis was intraprocedural reocclusion. The primary outcome for Step 2 of the analysis was delayed reocclusion. The secondary outcomes included 3-month favorable outcome defined by a modified Rankin score (mRS) of 0–2, and symptomatic intracranial hemorrhage (sICH), defined as any type of hemorrhage associated with an increase in the National Institutes of Health Stroke Scale (NIHSS) score by ≥4 points within 72 h.

### 2.8. Statistical Analysis

Statistical analysis was performed using the SPSS statistical package (version 20.0, Chicago, IL, USA). For the comparison between two groups, a χ^2^ test was performed for categorical variables, Student’s *t*-test was performed for continuous variables with a normal distribution, and Mann–Whitney’s U-test was performed for continuous variables without a normal distribution or ordinal variables. ANOVA was performed for the comparison among three groups, followed by Bonferroni correction for pairwise comparison.

## 3. Results

### 3.1. Patients

This study included 855 acute anterior circulation stroke patients, of whom 394 had an occlusion on the middle cerebral artery. Of the 394 patients, 107 were included with a diagnosis of ICAS, and 103 out of the 107 patients were further included with successful reperfusion after EVT. Next, 17 out of the 103 patients were excluded due to partial baseline occlusion (N = 12), no follow-up imaging (N = 3), or poor imaging quality (N = 2). Therefore, a total of 86 patients were selected for the study. All patients received tirofiban treatment. No patients received endarterectomy, which is one treatment for carotid artery stenosis [15] in this cohort. The patient selection process is detailed in Figure 1 and a detailed decision-making flow chart is depicted in Figure 2.

### 3.2. ICAS Classification Validation

Twenty-two patients underwent high-resolution magnetic resonance imaging, and acentric plaques could be observed in all these cases, which confirmed the diagnosis of ICAS.

#### 3.2.1. Step 1 Analysis: Intraprocedural Reocclusion

The intraprocedural reocclusion rate of the whole cohort was 7.0% (6/86), with a significant difference among the three groups (*p* = 0.003, Table 1). Pairwise comparison showed that for the patients with successful thrombectomy reperfusion, those with severe residual stenosis showed no difference from those with mild to moderate residual stenosis in terms of the intraprocedural reocclusion rate (Group 2a vs. Group 1a, 0/30 vs. 0/25, *p* = 1.0); on the other hand, for patients with severe residual stenosis, those with successful thrombectomy reperfusion showed a significantly lower intraprocedural reocclusion rate compared with no successful thrombectomy reperfusion (Group 2a vs. Group 3a, 0/30 vs. 6/31, *p* = 0.007).

The clinical and angiography characteristics of the three groups are summarized in Table 1. The baseline characteristics were similar among the three groups except for onset-to-groin puncture time. Group 3a (patients with severe residual stenosis but successful thrombectomy reperfusion) had more patients receiving thrombectomy beyond 8 h of onset compared with the other two groups (61.3% vs. 33.3% vs. 28%, *p* = 0.022).

#### 3.2.2. Step 2 Analysis: Delayed Occlusion

Moreover, for the group of patients with no successful thrombectomy reperfusion (*N* = 31, Group 3a from Step 1), the intraprocedural reocclusion rate was 19.4% (6/31); however, after successful reperfusion with the rescue treatment, this group of patients had a lower intraprocedural reocclusion rate (0/31 vs. 6/31, *p* = 0.032).

At the end of the EVT procedure, after rescue treatment of angioplasty and/or a stent, all patients achieved successful reperfusion after the EVT procedure. For the whole cohort, the delayed reocclusion rate was 2.3% (2/86), the rate of sICH was 1.2% (1/86), and the rate of good prognosis was 66.3% (57/86).

Sixty-one patients had mild to moderate residual stenosis (Group 1b), whereas 25 patients had severe residual stenosis (Group 2b, a typical case is illustrated in Figure 3). The two groups showed no significant difference regarding the reocclusion rate on follow-up angioplasty (0/61 vs. 2/25, *p* = 0.082). The two groups showed no significant difference in sICH (0/61 vs. 1/25, *p* = 0.298) or favorable functional outcome prognosis rate (41/61 vs. 16/25, *p* = 0.808, Table 2). The clinical and angiography characteristics of the three groups are summarized in Table 2.

### 3.3. Subgroup Analysis

In addition, in the group of patients who had successful thrombectomy reperfusion but severe stenosis (*n* = 30, Group 2a from Step 1), 13 patients received the rescue treatment and 12 out of the 13 patients resulted in mild to moderate residual stenosis afterwards, whereas 17 patients did not receive any rescue treatment and remained with severe residual stenosis after the EVT procedure. If we compare the 12 patients with residual stenosis management with the 17 patients without any management, the delayed reocclusion rate showed no significant difference (0/12 vs. 2/17, *p* = 0.510) and the favorable functional outcome rate showed no significant difference (6/12 vs. 12/17, *p* = 0.461).

## 4. Discussion

The main finding of this study is that reocclusion of the treated artery is fairly low in MCA ICAS-related occlusion patients with successful reperfusion through endovascular therapy, even with a high degree of residual stenosis after endovascular therapy. The reocclusion occurrence of ICAS patients is related to unsuccessful reperfusion rather than the severity of residual stenosis.

The findings of this study are consistent with a previous study on a Korean population [16]. In the previous study, within ICAS patients, the reocclusion rate was not found to be related to residual stenosis severity, but was influenced by reperfusion status. For patients with successful reperfusion, the reocclusion rate was much lower than that for patients without successful reperfusion (6.6% vs. 30.8%). In addition, for patients with residual stenosis, both the Korean study and this study showed a very much lower reocclusion rate after successful reperfusion. One possible explanation is the use of tirofiban. Studies [5,17] have shown that tirofiban, when delivered intra-arterially during a procedure, could dramatically reduce instant reocclusion during an endovascular procedure (85.7% reduction) as well as delayed reclusion on follow-up imaging (>70% reduction). Compared with the previous study, our study shows an even lower reocclusion rate with successful reperfusion. In this study, the tirofiban treatment was intravenously administered earlier, before the endovascular procedure, and its antiplatelet efficacy might be further improved [17]. The intravenous administration of tirofiban might also explain the low intracranial hemorrhagic rate in this study [18]. In summary, this study indicates that intravenous tirofiban treatment might be an effective treatment to prevent reocclusion and improve the success rate of endovascular treatment in ICAS patients.

The findings of our study support the use of rescue strategies, including stenting and/or angioplasty, in order to achieve successful reperfusion. However, stenting or angioplasty should be performed with caution if the aim is to address residual stenosis. It is still controversial whether stenting or angioplasty should be performed to address residual stenosis, according to the findings from previous studies [19,20,21]. The results of this study indicate that when successful reperfusion has been achieved, it might not be necessary to further perform stenting or angioplasty, even for patients with severe residual stenosis. It might not bring benefits but cause the following harms instead: (1) the procedure-related complication rate might increase with more endovascular operations [21,22], including the increased risk of perforating branch occlusions through emergent angioplasty; (2) the intracranial hemorrhage rate might increase with more endovascular operations [21,22], resulting from ischemic and reperfusion injuries to brain tissues [23,24], also result from intensive antiplatelet therapy after acute cerebral ischemia [25]. Furthermore, emergent angioplasty does not lower the reocclusion rate of the recanalized vessel [16,26,27,28] and does not improve patients’ functional outcome [29]. Certainly, in light of the high reocclusion rate in the context of acute stroke with ICAS, especially in cases of high residual stenosis, we should take sufficient time to observe the blood flow changes in the target artery. In our study, the observation time was no shorter than 20 min.

The limitations of our study include the following. First, this is a retrospective single-center study with relatively small number of patients. The baseline characteristics were balanced among groups, except for the onset-to-groin puncture time. The onset-to-puncture time might be related to reocclusion rate. This confounding factor needs to be further explored in future studies. Second, the definition of ICAS might include patients with residual stenosis due to dissection or residual thrombi. Therefore, in this study, high-resolution magnetic resonance imaging was performed to confirm the diagnosis of ICAS. However, only a subgroup of patients underwent high-resolution magnetic resonance imaging. Third, the findings of this study are probably more applicable to an Asian population with a high prevalence of ICAS and are limited to middle cerebral artery occlusion.

## 5. Conclusions

For acute ischemic stroke resulting from intracranial artery ICAS-related occlusion, endovascular treatment should focus on increasing successful reperfusion rather than recanalization of residual stenosis.

## Figures and Tables

**Figure 1 brainsci-12-00966-f001:**
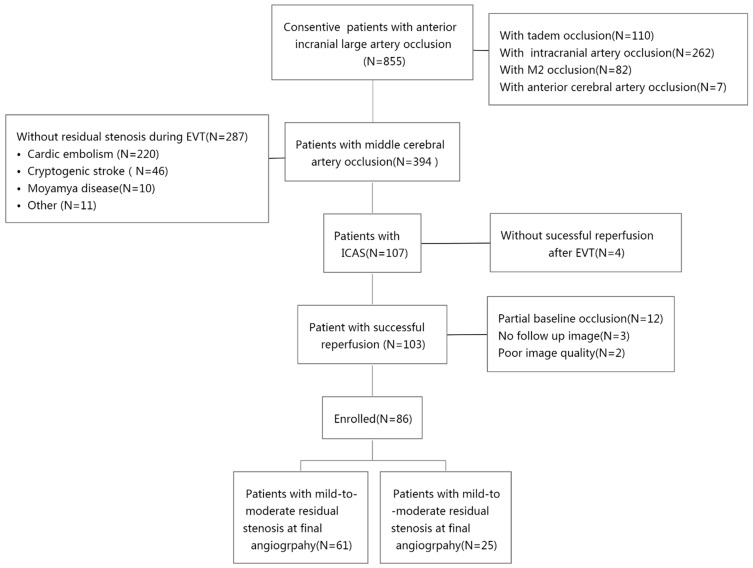
Patient selection and exclusion flow chart.

**Figure 2 brainsci-12-00966-f002:**
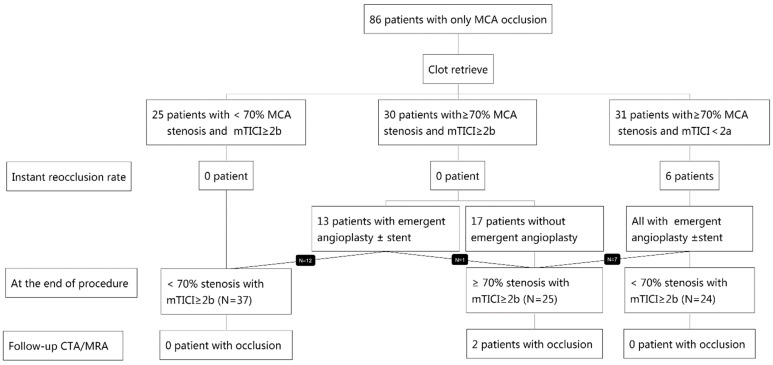
Decision-making flow chart. EVT, endovascular therapy; MCA, middle cerebral artery; mTICI, modified thrombolysis in cerebral infarction; CTA, computed tomography angiography; MRA, magnetic resonance angiography.

**Figure 3 brainsci-12-00966-f003:**
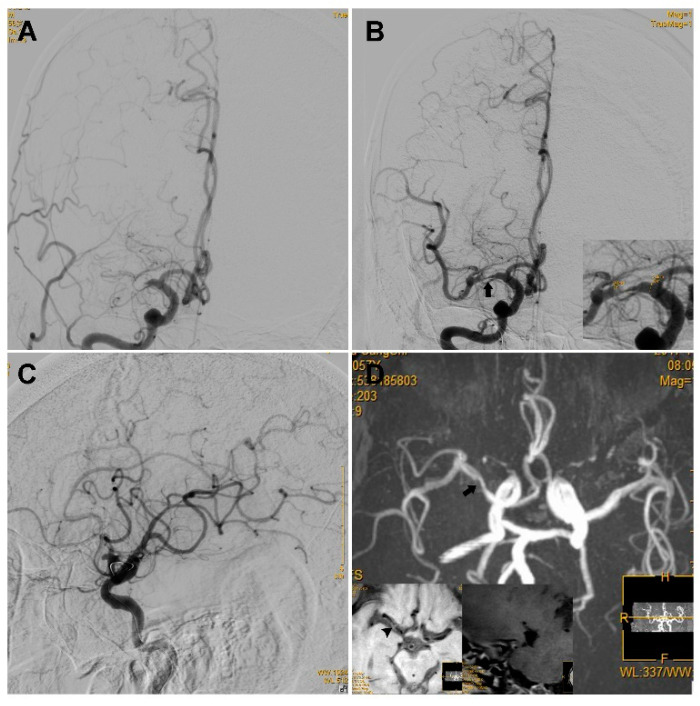
An ICAS case with a high degree of residual stenosis. An elderly patient presented with left limb weakness for 16 h, and the NIHSS score was 14. (**A**) The first run of DSA showed occlusion of the right MCA. (**B**) The anterior–posterior view of DSA showed a high degree of stenosis (black arrow) located at the right MCA after one pass of stent retrieval and emergent angioplasty via a 2.0–15 mm Maverick balloon; as the figure on the bottom right shows, the residual stenosis was 81.2% according to the WASID criteria. (**C**) Lateral view of the DSA showed that reperfusion with an mTICI of ≥2b was achieved and it was maintained for more than 20 min. (**D**). TOF-MRA performed 6 days after the operation showed a high degree of stenosis in the right MCA (black arrow). The high-resolution MRI scan on the bottom left showed an eccentric plaque (arrowhead) with enhanced stenosis that was observed at the right MCA. ICAS, intracranial atherosclerosis; NIHSS, National Institutes of Health Stroke Scale; DSA, digital subtraction angiography; MCA, middle cerebral artery; mTICI, modified thrombolysis in cerebral infarction; TOF-MRA, time of flight for magnetic resonance angiography; MRI, magnetic resonance imaging.

**Table 1 brainsci-12-00966-t001:** Baseline characteristics and clinical outcomes of patients in Step 1 of the analysis.

	Group 1a: mTICI ≥ 2b + Stenosis < 70% (*N* = 25)	Group 2a: mTICI ≥ 2b + Stenosis ≥ 70% (*N* = 30)	Group 3a: mTICI < 2b + Stenosis ≥ 70% (*N* = 31)	*p*-Value
Male Sex, *N* (%)	18 (72%)	22 (73.3%)	22 (71.0%)	0.979
Age (mean, years)	64 ± 15	66 ± 10	62 ± 12	0.497
Smoker	10 (40%)	16 (51.6%)	17 (56.7%)	0.457
Hypertension, *N* (%)	16 (72.7%)	24 (80.0%)	26 (86.7%)	0.454
DM *N* (%)	7 (28.0%)	8 (26.7%)	9 (29.0%)	0.979
Atrial fibrillation *N* (%)	0 (0%)	1 (3.3%)	0 (0%)	0.389
TIA *N* (%)	1 (4.0%)	1 (3.3%)	0 (0%)	0.554
Admission NIHSS (median, IQR)	14 (11,18)	13(11,17)	14 (11,18)	0.332
Onset-to-puncture time *N* (%)				0.022
Within 8 h	18 (72%)	20 (66.7%)	12 (38.7%)	
8–24 h	7 (28%)	10 (33.3%)	19 (61.3%)	
Good collateral flow, *N* (%)				
ASITN ≥ 3	14 (56.0%)	24 (77.4%)	15 (50.0%)	0.070
Instant reocclusion	0 (0%)	0 (0%)	6 (19.4%)	0.03

mTICI, modified thrombolysis in cerebral infarction; TIA, transient ischemic attack; DM, diabetes mellitus; NIHSS, National Institutes of Health Stroke Scale; IQR, interquartile range; ASITN, American Society of Interventional and Therapeutic Neuroradiology collateral grading system.

**Table 2 brainsci-12-00966-t002:** Baseline characteristics and clinical outcomes of patients in Step 2 of the analysis.

	Group 1b: mTICI ≥ 2b + Stenosis < 70% (*N* = 61)	Group 2b: mTICI ≥ 2b + Stenosis ≥ 70% (*N* = 25)	*p*-Value
Male Sex N (%)	45 (73.8%)	17 (68.0%)	0.588
Age (mean, years)	63 ± 13	65 ± 12	0.434
Smoker N (%)	30 (50.8%)	12 (48.0%)	0.811
Hypertension, N (%)	45 (78.9%)	21 (84.0%)	0.595
DM N (%)	17 (29.3%)	7 (28.0%)	0.904
Atrial fibrillation N (%)	0 (0%)	1 (4.0%)	0.291
TIA N (%)	1 (1.6%)	1 (4.0%)	0.499
Admission NIHSS (median, IQR)	14 (11,17)	14 (10,20)	0.681
Onset-to-puncture time N (%)			0.982
Within 8 h	35 (57.4%)	15 (60.0%)	
8–24 h	26 (42.6%)	10 (40.0%)	
Good collateral flow, N (%)			0.627
ASITN ≥ 3	35 (58.3%)	16 (64.0%)	
sICH N (%)	0 (0%)	1 (4.0%)	0.298
Good prognosis N (%)	41 (67.2%)	16 (64.0%)	0.805
Mortality N (%)	0 (0%)	1 (4.0%)	0.291
Delayed reocclusion N (%)	0 (0%)	2 (8.0%)	0.082

mTICI, modified thrombolysis in cerebral infarction; DM, diabetes mellitus; TIA, transient ischemic attack; NIHSS, National Institutes of Health Stroke Scale; IQR, interquartile range; ASITN, American Society of Interventional and Therapeutic Neuroradiology collateral grading system; sICH, symptomatic intracranial hemorrhage.

## Data Availability

Not applicable.

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
