# Peer review of "Endovascular Treatment of ICAS Patients: Targeting Reperfusion Rather than Residual Stenosis"

_brainsci, 2022, doi:10.3390/brainsci12080966_

Round 1

Reviewer 1 Report

This is a retrospective single-center study investigating the incidence of re-occlusion among 86 stroke patients with ICAS affecting MCA. The authors describe a lower incidence of re-occlusion among those with more complete reperfusion. 

The study's topic is interesting and confirms prior observations from a Korean cohort (referenced by the investigators). 

However, it would add to the value of the report if the following comments are addressed:

- It is still unclear to me how ICAS was suspected prior to EVT which prompted the use of tirofiban prior to attempting thrombectomy. This is important to understand the study's results. In my practice, it is not uncommon for re-occlusion to happen even after successful reperfusion when residual high-grade stenosis is present. 

- In addition, the details of post-EVT medical treatment need to be clarified for those who underwent stenting and those who did not. 

- Reocclusion were evaluated by 2 independent raters but it is unclear if they also scored the degree of residual stenosis, and the reperfusion scores. This needs to be clarified in the Methods.

Author Response

This is a retrospective single-center study investigating the incidence of re-occlusion among 86 stroke patients with ICAS affecting MCA. The authors describe a lower incidence of re-occlusion among those with more complete reperfusion. 

The study's topic is interesting and confirms prior observations from a Korean cohort (referenced by the investigators). 

However, it would add to the value of the report if the following comments are addressed:

- It is still unclear to me how ICAS was suspected prior to EVT which prompted the use of tirofiban prior to attempting thrombectomy.

Reply: We have addressed the reviewer’s concern by clarifying the definition of ICAS in the manuscript as follows:

Page 3, line 107-109: “ICAS was suspected once the following signs were observed on the first run of DSA, the appearance of tapered sign [7] or/and significant fixed focal stenosis (>50%) at the site of occlusion during endovascular treatment [8] or/and phenomenon of “microcrater first pass effect” [9] or/and positive stent-unsheathed effect[10].”

Reference

  1. Liang, W.; Wang, Y.; Du, Z.; Mang, J.; Wang, J. Intraprocedural Angiographic Signs Observed During Endovascular Thrombectomy in Patients With Acute Ischemic Stroke: A Systematic Review. Neurology 2021, 96, 1080–1090.
  2. Lee, J.S.; Hong, J.M.; Lee, K.S.; Suh, H. Il; Demchuk, A.M.; Hwang, Y.-H.; Kim, B.M.; Kim, J.S. Endovascular Therapy of Cerebral Arterial Occlusions: Intracranial Atherosclerosis versus Embolism. J. Stroke Cerebrovasc. Dis. 2015, 24, 2074–2080.
  3. Chen, W.; Wu, Y.; Zhang, M.; Zhan, A.; Chen, Y.; Wu, Z.; Shi, Y.; Chen, B. Microcatheter “First-Pass Effect” Predicts Acute Intracranial Artery Atherosclerotic Disease-Related Occlusion. Neurosurgery 2018, 0, 1–10.
  4. Chen, W. huo; Yi, T. yu; Zhan, A. lai; Wu, Y.M.; Lu, Y. yu; Li, Y. min; Pan, Z. nan; Lin, D. lai; Lin, X. hui Stent-Unsheathed Effect Predicts Acute Distal Middle Cerebral Artery Atherosclerotic Disease-Related Occlusion. J. Neurol. Sci. 2020, 416.

This is important to understand the study's results. In my practice, it is not uncommon for re-occlusion to happen even after successful reperfusion when residual high-grade stenosis is present. 

Reply: We have taken the reviewer’s advice by adding the following explanation of low re-occlusion rate in the discussion:

Page 8, line 237-242: “In addition, for patients with residual stenosis, both the Korean study and this study showed a very low re-occlusion rate after successful reperfusion. One possible explanation is the use of Tirofiban. Studies[4,15] have shown that Tirofiban, when delivered intra-arterially during procedure, could dramatically reduce instant re-occlusion during endovascular procedure (85.7% reduction) as well as delay reclusion on follow-up imaging (>70% reduction).””

Reference:

4.Kang, D.-H.; Kim, Y.-W.; Hwang, Y.-H.; Park, S.-P.; Kim, Y.-S.; Baik, S.K. Instant Reocclusion Following Mechanical Thrombectomy of in Situ Thromboocclusion and the Role of Low-Dose Intra-Arterial Tirofiban. Cerebrovasc. Dis. 2014, 37, 350–355.

  1. Yan, Z.; Shi, Z.; Wang, Y.; Zhang, C.; Cao, J.; Ding, C.; Qu, M.; Xia, Y.; Cai, J.; Zhang, X.; et al. Efficacy and Safety of Low-Dose Tirofiban for Acute Intracranial Atherosclerotic Stenosis Related Occlusion with Residual Stenosis after Endovascular Treatment. J. Stroke Cerebrovasc. Dis. 2020, 29, 1–6.

- In addition, the details of post-EVT medical treatment need to be clarified for those who underwent stenting and those who did not. 

Reply: We have addressed the reviewer’s concern about the details of post-EVT medical treatment for those who underwent stenting and those who did not in the manuscript as follows:

Page 2-3, line 90-98. A loading dose (10 μg/kg) of a glycoprotein IIb/IIIa inhibitor (Tirofiban) was administered intravenously for 3 minutes once ICAS was considered as the cause of stroke during EVT procedure (prior to attempting thrombectomy); then, an infusion of tirofiban at 0.1-0.15 μg/kg/minute was administered, and this infusion was continued for 12-36 hours after the EVT operation [6]. If no brain hemorrhages were detected by the CT scan performed at least 12 hours after the operation, a loading dose of aspirin (100 mg/day) plus clopidogrel (300 mg/day), followed by a dose of aspirin (100 mg/day) plus clopidogrel (75 mg/day) for at least 3 months was administered in both patients who received stenting or not.

Refference:

[6].Yi, T.-Y.; Chen, W.-H.; Wu, Y.-M.; Zhang, M.-F.; Chen, Y.-H.; Wu, Z.-Z.; Shi, Y.-C.; Chen, B.-L. Special Endovascular Treatment for Acute Large Artery Occlusion Resulting From Atherosclerotic Disease. World Neurosurg. 2017, 103

- Reocclusion were evaluated by 2 independent raters but it is unclear if they also scored the degree of residual stenosis, and the reperfusion scores. This needs to be clarified in the Methods.

Reply: We have addressed the reviewer’s concern by adding the following information in the manuscript:

Page 3, line 111-112. All angiographic classifications, including the grade of collateral flow, the degree of residual stenosis, and the reperfusion score, were evaluated by the two independent raters who rated re-occlusion status (Z.A.L. and L.Y.M). Interobserver disagreements were resolved by consensus.

Reviewer 2 Report

Congratulations for your  extensive study;  required changes needed

Reviewer 3 Report

The current paper from Yi et al  studies the reoocclusions in ICAS patients.

The paper is well written. It is a retrospective study from a wide prospective database. 

I have the following queries:

  • Follow up is not too long. No patients in this database, treated after 2019, was included in the presented analysis. Why? Is it possible that no patients treated later than 2019 was eligible for this analysis?
  • I’ve really appreciated the use of tirofiban, but I haven’t seen informations about antithrombotic therapy used later than tirofiban administration. Please specify.
  • Is it possible that differences in reocclusion rate in group 3a can be due to an higher onset-to-puncture time? Please address this point
  • Please fix grammar and some mistakes during the text
  •  

Author Response

Reviewer Three:

The current paper from Yi et al  studies the reoocclusions in ICAS patients.

The paper is well written. It is a retrospective study from a wide prospective database. 

I have the following queries:

  • Follow up is not too long. No patients in this database, treated after 2019, was included in the presented analysis. Why? Is it possible that no patients treated later than 2019 was eligible for this analysis?

Reply: Regarding the follow-up time of individual patient, in this study, we used 90 days as the end of follow up, since the 90-days modified Rankin Score is one of classic follow-up clinical outcome in stroke study.

Regarding the study period, since 2019, our center starts to lead and participate in multiple clinical trials, including a Tirofiban trial. Most thrombectomy patients have been enrolled in ongoing trials since 2019 and excluded from this study.

  • I’ve really appreciated the use of tirofiban, but I haven’t seen informations about antithrombotic therapy used later than tirofiban administration. Please specify.

Reply: have addressed the reviewer’s concern by adding the following information in the manuscript as follows:

Page 2, line 90-97. A loading dose (10 μg/kg) of a glycoprotein IIb/IIIa inhibitor (Tirofiban) was administered intravenously for 3 minutes once ICAS was considered as the cause of stroke during EVT procedure (prior to attempting thrombectomy); then, an infusion of tirofiban at 0.1-0.15 μg/kg/minute was administered, and this infusion was continued for 12-36 hours after the EVT operation [6]. If no brain hemorrhages were detected by the CT scan performed at least 12 hours after the operation, a loading dose of aspirin (100 mg/day) plus clopidogrel (300 mg/day), followed by a dose of aspirin (100 mg/day) plus clopidogrel (75 mg/day) for at least 3 months was administered in both patients who received stenting or not.

  • Is it possible that differences in re-occlusion rate in group 3a can be due to a higher onset-to-puncture time? Please address this point

Reply: We have added the following information to address the reviewer’s concern :

Page 6, line 174-177, The baseline characteristics were similar among the 3 group except for onset-to-groin time. Group 3a, patients with severe residual stenosis but successful thrombectomy reperfusion, had more patients receiving thrombectomy beyond 8 hours of onset compared to the other two groups (61.3% vs. 33.3% vs. 28%, p=0.022).Page 10, line 290-294: “The limitations of our study include the following. First, this is a retrospective, single center study with relatively small number of patients. Baseline characteristics were balanced among groups, except for the onset-to-groin time. The onset-to-puncture time might be related to re-occlusion rate. Such confounding factor needs to be further explored in future studies.”

  • Please fix grammar and some mistakes during the text
  • Reply: We have sent the manuscript to senior co-authors for proof reading. The grammar mistakes and typos have been corrected and marked in blue colour.

Reviewer 4 Report

Dear authors,

The introduction is a very important part of an article: "The introduction should briefly place the study in a broad context and highlight why it is important. It should define the purpose of the work and its significance. The current state of the research field should be carefully reviewed and key publications cited. Please highlight controversial and diverging hypotheses when necessary. Finally, briefly mention the main aim of the work and highlight the principal conclusions.

The reference numbers should be placed in square brackets [ ].

The overall manuscript needs some ameliorations.

Author Response

Dear authors,

The introduction is a very important part of an article: "The introduction should briefly place the study in a broad context and highlight why it is important. It should define the purpose of the work and its significance. The current state of the research field should be carefully reviewed and key publications cited. Please highlight controversial and diverging hypotheses when necessary. Finally, briefly mention the main aim of the work and highlight the principal conclusions.

Reply: We have taken the reviewer’s advice by updating the introduction as follows:

Page 1,line 35-53. T

Endovascular therapy (EVT) has become a routine practice for acute ischemic stroke caused by large vessel occlusion in highly specialized centers with dedicated Stroke units [1]. However, EVT procedure does not always lead to good clinical outcomes. One of the reasons is the re-occlusion of targeted artery after procedure. The incidence of postoperative acute re-occlusion of treated arteries has been reported range from 3%-9%[2], and the incidence has been shown to be higher in cases of intracranial atherosclerosis (ICAS)-related occlusion, especially in cases with high residual stenosis[3], because insufficient blood flow caused by high residual stenosis through the target artery lead to acute thrombus formation[3–5].Therefore, if there is sufficient blood flow through the occluded artery affected by ICAS, the incidence of acute thrombosis after endovascular therapy can be relatively low. Furthermore, in our clinical practice, we observed unsuccessful reperfusion as defined by mTICI<2b rather than residual stenosis severity was related to artery re-occlusion in ICAS patients. We hypothesized that the incidence of acute target arterial re-occlusion is low if mTICI≥2b reperfusion is achieved in patients with MCA ICAS-related occlusion even if with high degree of residual stenosis after endovascular therapy. Therefore, our study aimed to assessed whether re-occlusion was determined by reperfusion status or residual stenosis severity in ICAS patients.

Reference

  1. Powers, W.J.; Rabinstein, A.A.; Ackerson, T.; Adeoye, O.M.; Bambakidis, N.C.; Becker, K.; Biller, J.; Brown, M.; Demaerschalk, B.M.; Hoh, B.; et al. 2018 Guidelines for the Early Management of Patients With Acute Ischemic Stroke: A Guideline for Healthcare Professionals From the American Heart Association/American Stroke Association; 2018; Vol. 49; ISBN 0000000000000.
  2. Mosimann, P.J.; Kaesmacher, J.; Gautschi, D.; Bellwald, S.; Panos, L.; Piechowiak, E.; Dobrocky, T.; Zibold, F.; Mordasini, P.; El-Koussy, M.; et al. Predictors of Unexpected Early Reocclusion after Successful Mechanical Thrombectomy in Acute Ischemic Stroke Patients. Stroke 2018, 49, 2643–2651, doi:10.1161/STROKEAHA.118.021685.
  3. Kim, G.E.; Yoon, W.; Kim, S.K.; Kim, B.C.; Heo, T.W.; Baek, B.H.; Lee, Y.Y.; Yim, N.Y. Incidence and Clinical Significance of Acute Reocclusion after Emergent Angioplasty or Stenting for Underlying Intracranial Stenosis in Patients with Acute Stroke. Am. J. Neuroradiol. 2016, 37, 1690–1695, doi:10.3174/ajnr.A4770.
  4. Kang, D.-H.; Kim, Y.-W.; Hwang, Y.-H.; Park, S.-P.; Kim, Y.-S.; Baik, S.K. Instant Reocclusion Following Mechanical Thrombectomy of in Situ Thromboocclusion and the Role of Low-Dose Intra-Arterial Tirofiban. Cerebrovasc. Dis. 2014, 37, 350–355, doi:10.1159/000362435.
  5. Heo, J.H.; Lee, K.Y.; Kim, S.H.; Kim, D.I. Immediate Reocclusion Following a Successful Thrombolysis in Acute Stroke: A Pilot Study. Neurology 2003, 60, 1684–1687.

The reference numbers should be placed in square brackets [ ].

Reply: We have placed all reference numbers in square brackets [ ].

The overall manuscript needs some ameliorations.

Reply: We have sent the manuscript to senior co-authors for proof reading. The grammar mistakes and typos have been corrected and marked in blue colour. We have updated the manuscript according to all reviewers’ feedbacks for clarification and improvement.

Round 2

Reviewer 1 Report

Thank you.  I have no further comments. 

Reviewer 4 Report

Dear Authors,

I still find the introduction quite short. Even so, the work has been ameliorated.

kind regards